# Electrolyzed Water and Its Pharmacological Activities: A Mini-Review

**DOI:** 10.3390/molecules27041222

**Published:** 2022-02-11

**Authors:** Bo-Kai Chen, Chin-Kun Wang

**Affiliations:** Department of Nutrition, Chung Shan Medical University, 110, Section 1, Jianguo North Road, Taichung 40201, Taiwan; gargon88@gmail.com

**Keywords:** electrolyzed water, disinfecting agent, infectious diseases, chronic diseases

## Abstract

Electrolyzed water (EW) is a new type of cleaning and disinfecting agent obtained by means of electrolysis with a dilute sodium chloride solution. It has low cost and harm to the human body and is also friendly to the environment. The anode produces acidic electrolyzed water (AEW), which is mainly used to inhibit bacterial growth and disinfect. The cathode provides basic electrolyzed water (BEW), which is implemented to promote human health. EW is a powerful multifunctional antibacterial agent with a wide range of applications in the medicine, agriculture, and food industry. Studies in vitro and in vivo show that it has an inhibitory effect on pathogenic bacteria and viruses. Therefore, EW is used to prevent chronic diseases, while it has been found to be effective against various kinds of infectious viruses. Animal experiments and clinical trials clearly showed that it accelerates wound healing, and has positive effects in oral health care, anti-obesity, lowering blood sugar, anti-cancer and anti-infectious viral diseases. This review article summarizes the application of EW in treating bacteria and viruses, the prevention of chronic diseases, and health promotion.

## 1. Introduction

The aging of populations has been increasing recently, and is projected to increase from 11% to 22% by 2050. The health issues of the elderly could greatly influence society and economics. Healthy lifestyles and high life expectancy have become very critical, especially in highly aging areas. Many studies show that many elderly people over 100 years old live in areas with clean air and water, especially with a high quality of water. Researchers have used electrolyzed technology to produce multi-functional EW. EW shows many kinds of benefits. Acidic EW effectively suppresses many harmful bacteria [1], and has also been used as a new type of disinfectant (containing HOCl) and cleaning agent (containing NaOH) in recent years [2]. EW is also used in combination with other methods, such as organic acids, ultrasound [3] and mild heating [4] to perform more effective sterilization. AEW is more effective as a disinfectant. BEW is mostly used as drinking water to promote health benefits and prevent some diseases [5,6,7,8].

Severe acute respiratory syndrome coronavirus 2 (SARS-CoV-2) was discovered at the end of 2019, and was declared a pandemic in early 2020. It is highly transmissible, and has spread around the world. Vaccination is one method of controlling this disease. However, new variants, including delta and omicron, are still very highly transmissible and continue to spread. Other therapy or prevention techniques are needed to protect public health. The Centers for Disease Control and Prevention (CDC) recommends that public health practitioners and organizations prioritize prevention strategies for indoor environments. No single strategy is sufficient to prevent transmission, and multiple interventions should be used simultaneously to reduce the spread of the disease [9]. In addition to vaccination, proven strategies to combat SARS-CoV-2 transmission include the proper use of masks at all times [10,11], maximizing ventilation through dilution [12,13] and air filtration [14], physical distancing and avoiding crowds [15,16]. Washing hands and regular cleaning frequently touched surfaces should also be encouraged.

Effective disinfectants to kill pathogens and disrupt biofilm formation in the environment and promote human health is one of the most important strategies to combat infectious diseases. However, some techniques have disadvantages, such as high cost, low efficacy, the problem of residual chemicals, and side-effects including irritation to human skin [17,18]. As a practical application, a disinfectant should demonstrate high antibacterial efficacy and be non-toxic to humans [19].

Therefore, the use of EW is a good option. EW has the advantages of low cost and low harm to the human body. AEW exhibits strong bactericidal ability and antiviral potential, while BEW has multiple functions to promote human health. The purpose of this review is to introduce the latest developments in electrolyzed water and provide new perspectives in the clinical fields. Additionally, this study clearly introduces EW, including its physicochemical properties, history, production, antimicrobial effects, its preventative effects on diseases and health promotion.

## 2. History of Electrolyzed Water

The concept of EW originated in Japan and is currently recognized by various developed countries as the safest and most advanced form of water. When tap water passes through electrodialysis, the water is electrolyzed. As early as 1931, Japan developed the world’s first electrolyzed water generator to adjust the pH of water, and then began to study the effects of EW on animals and plants. After more than 20 years of research on the effects of EW on the growth of animals and plants, it was confirmed to effectively promote the growth and development of animals and plants, and also to prevent and treat certain human diseases. In 1966, the Ministry of Health and Welfare of Japan officially approved drinking electrolyzed water, the water device was used as a medical device. In 1994, Japan established the Functional Water Research Committee, a third-party scientific research organization composed of medical, agricultural, and engineering experts to conduct further research on electrolyzed water. After 7 years of large-sample, multi-center, randomized double-blind human clinical experiments, in March 1999 the Japanese Ministry of Health and Welfare reconfirmed the therapeutic effect of basic electrolyzed water, and electrolyzed functional water finally entered medical academia, leading to an era of extensive research, recognition and application. After that, the EW was introduced into the markets of various countries, and has been greatly developed in Europe, the United States, Japan, South Korea, Taiwan, and Southeast Asia.

## 3. Generation and Classification of Electrolyzed Water

EW is the product of the electrolysis of a dilute NaCl or KCl-MgCl_2_ solution in an electrolysis cell, within which a diaphragm (septum or membrane) separates the anode and cathode. During electrolysis, NaCl dissolved in deionized water dissociates into negatively charged chlorine (Cl^−^) and positively charged sodium (Na^+^). At the same time, hydroxide (OH^−^) and hydrogen (H^+^) ions are formed. Negatively charged ions such as Cl^−^ and OH^−^ move to the anode to give up electrons and become oxygen gas (O_2_), chlorine gas (Cl_2_), hypochlorite ion (OCl^−^), hypochlorous acid (HOCl), and hydrochloric acid, and positively charged ions such as H^+^ and Na^+^ move to the cathode to take up electrons and become hydrogen gas (H_2_) and sodium hydroxide (NaOH) (Figure 1). EW is a chlorine-based disinfectant that can be relatively easily prepared on-site by electrolyzing a solution of pure table salt (sodium chloride, NaCl) using one of many pieces of commercially available electrolysis equipment [20]. With the latest developments in technology, the industry has improved this technology to increase the effectiveness of electronic warfare. Since 2010, many innovative companies have emerged in the market, and electronic warfare generators can also be used by individuals and small businesses [21,22]. EW has three main physical properties—the available chlorine concentration (ACC), the pH value and the oxidation-reduction potential (ORP)—and the difference in these properties will load the EW into different sterilization effects. Numerous studies have shown the interaction effects among these factors [23,24]. According to the different devices, electrolyte and electrolysis conditions, EW can be classified into the following categories: AEW, WAEW, NEW and BEW. The characteristics of EW are shown in Table 1.

### 3.1. Acidic Electrolyzed Water

The acidic electrolyzed water thus collected has a pH of about 2.2 to 2.7, the ORP is greater than 1100 mV, and the ACC is 20–60 ppm, and it can inactivate most pathogenic bacteria [25,26]. The main component is hypochlorous acid. AEW is a kind of medical product with a strong sterilization effect [27].

### 3.2. Weak Acidic Electrolyzed Water

Weak acidic electrolyzed water is also known as slightly acidic hypochlorous acid water and slightly acidic oxidizing potential water. The pH value is 5.0~6.5, the ORP is about 850 mV, and the ACC is 10–30 ppm. This electrolyzed water has a high bactericidal effect and is colorless and odorless. The original purpose of WAEW was to be a substitute for fungicides in food equipment. It will not affect the taste and aroma of the food and achieves sterilization, and the upper limit of ACC is 30 ppm. WAEW shows more stable sterilization than AEW [28].

### 3.3. Basic Electrolyzed Water

Basic electrolyzed water is generated from the cathode. The alkaline solution has a pH of 10 to 13, an ORP of −800 to −900 mV, and an ACC is 80–100 ppm [29]. BEW lacks strong sterilization, which limits its application in the food industry. Although BEW also has an antibacterial effect, its effect is not as strong as AEW. BEW is mainly explored for its effects on disease prevention or health promotion. Clinical practice shows that the intake of BEW greatly improves gastrointestinal symptoms [30], and also suppresses the growth of anaerobic bacteria in the oral cavity [31]. The clinical application of BEW on human body has great potential.

### 3.4. Neutral Electrolyzed Water

Neutral electrolyzed water is a new type of disinfectant, which replaces sodium disinfectants and is regarded as a mixed oxidant. Partially mixed hydroxide ions enter the cathode side, resulting in a near neutral pH of about 7.0–8.0, while the ORP is 700–900 mV and the ACC is 30–200 ppm [32]. In recent years, NEW has been a novel and potential disinfectant, owing to its lower cost and minimal harm to the human body when compared with AEW.

## 4. Factors Affecting the Antimicrobial Properties of Electrolyzed Water

There are three main factors that affect the antimicrobial efficacy of EW: ACC, pH value and ORP [33]. These three main factors of EW influence each other, and change with time and temperature. The pH value plays an important role in the formation of various chlorine substances. When the pH of the solution is 5.0 to 6.5, the primary chlorine is in the form of HOCl, and its disinfection ability is 80 times that of -OCl [34]. The results show that the ORP and ACC of EW decreased significantly when the pH increased from acid (pH 2.5) to alkaline (pH 9.0) [23]. In addition, EW generated at different temperatures shows different ACC [35]. ACC can be reduced with the increasing storage time [23]. On the other hand, free radicals (such as hydroxyl radicals (OH·)) are also considered to be bactericidal components of EW [36]. OH· radicals destroy the cell structure of the microorganism and the decontaminated function of NEW may be derived from OH· radicals [37].

## 5. Mechanism and Disinfective Effect of Electrolyzed Water

The disinfective effect of EW mainly depends on its low pH, high ORP, and the synergistic effect of HClO, Cl_2_, H_2_O_2_ and hydroxyl (OH^−^). A low pH will affect the permeability of cell membranes and prevent them from reproducing. High ORP will affect the metabolic compounds in bacterial cells and lead to cell death. OH^−^ and H_2_O_2_ can damage cell lipid membranes, denature proteins and prevent them from reproducing, destroying bacteria by cutting DNA to prevent enzyme activation [38,39]. At present, the sterilization of EW has not been fully elucidated, but a model to explain the sterilization mechanism of EW has been roughly developed. First, after EW treatment, the morphology of the cell surface is changed from smooth, continuous, and bright to rough, shrunken and even dissolved. At the same time, the bacterial protective barrier (cell wall and cell membrane) is attacked and destroyed by chlorine substances, which will increase the permeability of the cell membrane and the leakage of intracellular materials (K^+^, protein and DNA) [40,41].

## 6. The Function and Application of Acidic Electrolyzed Water

High ORP in EW can cause changes in metabolic flux and ATP production, which may be due to changes in electron flow in cells. A low pH may make the outer membrane of bacterial cells sensitive to the entry of HOCl into bacteria [42]. In recent years, AEW has been used as a disinfectant or cleaning agent in various industries, such as agriculture, livestock, medicine, and food.

### 6.1. Antimicrobial

Comparing the inhibitory effects of AEW and sterile deionized water containing free chlorine on pathogenic bacteria, the results reveal reductions in the bacterial counts of both pathogens similar to those observed with AEW [26]. In subsequent studies, the antibacterial effect of AEW was proved repeatedly [27,43,44]. In addition, the results show that the antibacterial effect of AEW significantly suppressed the survival of all bacteria, and that the survival of bacteria showed a negative correlation with ORP (r = −0.7158~−0.9982) and time (r = −0.8688~−1.0000) [45]. Comparing the disinfective effects of WAEW (ACC 33 mg/L, pH 6.4, ORP 834.9 mV), NaClO (ACC 30 mg/L, pH 10.83, ORP 304.7 mV) and 0.1% HCl (pH 1.93) on Staphylococcus aureus [25], it was found that the disinfective effect of WAEW treatment (reduction of 5.8 log CFU/mL) was significantly higher (*p* < 0.05) than that of the treatment with NaClO (reduction of 3.26 log CFU/mL) and HCl (reduction of 2.73 log CFU/mL), while its antibacterial effect is even better than AEW [46]. The antimicrobial activity of AEW and WAEW is shown in Table 2.

### 6.2. Combination Preservation Technologies

In recent years, the incidence of foodborne disease outbreaks has increased. Therefore, to reduce the occurrence of foodborne diseases, many disinfection methods, including chemical and physical treatments, have been employed. Studies have emphasized that EW is a novel disinfectant, and there are also many combined traditional disinfection methods with EW, such as physical heating treatment, ultrasound and the addition of chemical organic acids.

#### 6.2.1. Organic Acids

Organic acids are considered to have great potential to control a variety of microorganisms. In general, most organic acids (such as lactic acid, levulinic acid, citric acid, and fumaric acid) are safe, comply with the strict regulations of organic food, and show strong bactericidal effects on various pathogens, among which the disinfective effect depends on the pKa of the non-dissociated form and the hydrogen ion provided in the aqueous system [55,56]. Samples were treated in WAEW, AEW and WAWE + fumaric acid (FA) at 25, 40, and 60 °C for 1, 3, and 5 min, respectively, and the treated meat was air-packed and stored at 4 or 10 °C. The results show that the combined treatment of WAEW and FA (WAEW washing followed by FA washing) greatly improved the inhibitory effect [52]. The combinations of AEW and WAEW with organic acids are shown in Table 3.

#### 6.2.2. Ultrasound

Ultrasound (US) is a form of non-heating physical treatment. It is recognized as an emerging technology to improve the microbial quality and safety of fresh products. It has more advantages in energy saving, cost, reducing physical damage, maintaining fruit quality and improving shelf life. It is used as an antibacterial agent in food processing [58]. US enhanced the antibacterial effect of WAEW, reducing the total aerobic bacteria of cherry tomatoes and strawberries by 1.77 and 1.29 log, and reducing yeast and mold by 1.50 and 1.29 log, respectively [59]. The combinations of AEW and WAEW with ultrasound are shown in Table 4.

#### 6.2.3. Thermal Processing

Thermal processing is used to reduce the number of microorganisms and inactivate enzymes to extend the shelf life of the products. At present, the combination of EW and mild thermal processing (also known as mildly heated EW) has become effective in maintaining the quality and safety of fresh-cut and ready-to-eat organic agricultural products. Although AEW has a strong disinfective effect, the effect of reducing bacteria at room temperature is limited. AEW is used to treat live mussels and clams at 22 °C for 1 to 2 h. *Listeria monocytogenes* is reduced by only 1.0 and 1.1 log 10 CFU/g [49]. Increasing the temperature and exposure time of EW, the highest bacterial reduction rate was found at 65 °C [63]. Comparing the contents of *E. coli* and *Salmonella* after washing the lettuce with AEW twice and pre-washing with 20 °C BEW and then washing the lettuce with AEW, it was found that the latter group demonstrated a better antibacterial effect than the former one [64]. The combinations of AEW and WAEW with thermal processing are shown in Table 5.

#### 6.2.4. UVC-LED

UV irradiation (100–400 nm) has also been widely used in the food industry to ensure the microbiological safety of drinking water and various foods [66]. Lettuce washed with WAEW and then irradiated with UVC-LED demonstrated a better antibacterial effect. Compared with WAEW with an ACC of 40 ppm (1.0 log 10 CFU/g) and exposure for 7 min, WAEW with an ACC of 80 ppm resulted in a greater reduction in Salmonella (1.44 log 10 CFU/g) [67]. Not only was the impact of WAEW on *Salmonella* tested, but also that on *E. coli*. The results show that the changes in *Salmonella* were similar to those in *E. coli*. After treatment with 20, 40 and 60 ppm ACC of WAEW for 1 min, *Salmonella* and *E. coli* were significantly reduced (*p* < 0.05) [68]. The combinations of AEW and WAEW with UVC-LED are shown in Table 6.

### 6.3. Antifungal

AEW showed strong antifungal activity on *Aspergillus flavus* conidia and mycelium. The results show that both AEW and NEW significantly reduced the wet and dry weights of *Aspergillus flavus* mycelia in all cases when compared with the control [70]. The antifungal ability of HRM strains was evaluated when exposed to different reagents, including AEW and five liquid commercial disinfectants. The results show that efficacy increased when the concentration of EW was increased [71]. WAEW with ACC (102 mg/L, pH 3.9) produced by the saturated NaCl solution exhibited inhibitory effects on *P. digitatum*. It was proved that the ACC in WAEW is sufficient to reduce the growth of fungi and directly related to the contact time [72]. The antifungal effects of AEW and WAEW are shown in Table 7.

### 6.4. Antiviral

Human norovirus and hepatitis A virus (HAV) are representative food-borne viruses. Human noroviruses cause acute non-bacterial gastroenteritis and have been listed as the pathogens with the highest total cost of foodborne diseases in the United States [76]. AEW with 10 mg/L of ACC achieved a reduction of 0.74 log PFU/mL in murine norovirus (MNV-1) after 1 min of exposure. Additionally, the efficacy was increased with ACC. The results indicate that AEW had strong virucidal activity against MNV-1 and HAV, and that ORP, pH and ACC were important factors in the virucidal activity of AEW [77]. Evaluating the SARS-CoV-2 inactivation efficacy of AEW as an alternative disinfectant, the results show that the viral titer of an AEW-treated SARS-CoV-2 solution was below the detection limit (≥99.99% inactivation; decrease of ≥4.25 log_10_ TCID_50_/mL) [78]. Recent studies [79,80] showed the effect of EW against SARS-CoV-2, produced at concentrations as low as 100 ppm of HOCl, and a number of HOCl-based products have been developed in accordance with EPA guidelines for use as a SARS-CoV-2 standard [81]. The antiviral effects of AEW and WAEW are shown in Table 8.

### 6.5. Wound Healing

Severe burns can easily lead to sepsis due to wound infection, even if acute burn shock is well controlled. Wound healing is a complex, overlapping but systemic mechanism that is strongly controlled to restore the integrity of the skin. This process includes four stages, namely the hemostasis, inflammation, proliferation and remodeling stages. pH has a great influence on wound healing, because it can control wound infection, increase antibacterial activity, change the activity of proteases, such as matrix metalloproteinases (MMPs) and tissue inhibitors of MMPS (TIMPs), release oxygen, reduce the toxic end products of bacteria, and enhance epithelialization and angiogenesis [85].

AEW (pH = 2.65, ORP = 1159 mV, ACC = 32.1 ppm) was sprayed on the skin wounds of hairless mice 3 times a day for 7 days. The wound morphological and histological features and immune-redox markers were compared with the saline (Sal-) and alcohol (Alc-) treatment groups. The results show that the wound healing rate of the AEW group was significantly higher than that of the Sal group on the second, fourth, fifth, and sixth days [86]. Uninfected rats were assigned to the following groups: group 1, no irrigation (*n* = 6); group 2, rinsed with 20 mL of normal saline once a day (*n* = 6) 8 h after the removal of the eschar; and group 3, irrigation with 20 mL AEW (pH = 2.7, ACC = 50 ppb to 40 ppm, ORP = at least +1000 mV) once a day starting 8 h after the removal of the eschar (*n* = 6). The results show that one out of six rats in group 1 (epithelialization: 70 days), one out of six rats in group 2 (48 days), and five out of six rats in group 3 were observed to display complete epithelialization (58.4 ± 12.0 days)—this difference is significant. These results suggest that the use of AEW for irrigation and disinfection could prevent burn wound infection without inhibiting burn healing. Further clinical studies are needed to clarify the role of AEW irrigation in burn wounds [87].

Aiming to determine the effect and mechanism of SAEW (pH = 5–6.5, ORP = 800 mV, ACC = 25 ppm) on skin wounds in hairless mice, it was found that the SAEW treatment group showed the highest percentage of wound reduction (*p* < 0.01). The antioxidant activities of the SAEW group, such as glutathione peroxidase, catalase and myeloperoxidase activities, exceeded the total active oxygen in the skin. This study showed that SAEW was effective in wound healing in hairless mice through immune redox regulation and heals better than traditional drugs [88].

### 6.6. Anti-Obesity

Fourteen mice (three weeks old) were used in an experiment. They were divided into two groups—one group was given free AEW (pH ≤ 2.7, ORP ≥ 1100 mV, ACC = 20–60 ppm) as drinking water (test group), and the other was given free tap water as drinking water (control group). The experiment lasted for eight weeks. The results show that no significant difference was observed in the weight change between the control group and the test group. The gastrointestinal tracts (tongue, esophagus, stomach, jejunum, cecum, colon) were all normal, with no inflammation or other abnormalities. This suggests that AEW has no systemic effect, and is safe to use in mouthwashes, although more detailed research is needed [89].

## 7. The Function and Application of Basic Electrolyzed Water

Basic electrolyzed water was first developed in Japan, and its efficacy was explored in the medical and agricultural fields. BEW was recognized by Japan and South Korea as a new type of drug for the treatment of various intestinal diseases, because of its known efficacy. The health benefits associated with consuming BEW are based on the ability to neutralize and scavenge free radicals present in cells, which prevents oxidative damage to DNA, proteins and other molecules. Therefore, it may play an important role in improving different diseases such as cancer, diabetes and kidney damage [90]. BEW also has some potential benefits, such as improving digestive tract health, accelerating wound healing, oral health care, and anti-obesity. The following sections will summarize the application of EW in clinical research.

### 7.1. Type 2 Diabetes Mellitus (T2DM)

Type 2 diabetes mellitus (T2DM) is a metabolic disorder with multiple causes, represented by chronic hyperglycemia and carbohydrate, lipid and protein metabolism disorders associated with insulin resistance, the progressivity of which may also affect insulin secretion defects or combinations. Diabetes not only increases the prevalence of chronic diseases such as cardiovascular disease and nephropathy, but also increases the risk of infection and death [91]. Oxidative stress is related to the occurrence of T2DM through insulin resistance [92]. BEW is a kind of electrolytically treated water that can increase its reduction potential. It is a solution can provide a safe source of free electrons to prevent normal tissues from being oxidized by free radicals. Some studies have shown that hydrogen (H_2_) molecules show some therapeutic effects by acting as new antioxidants [93].

Dividing genetically diabetic male db/db mice (C57BL/6J db/db) into two groups, the control group and the intervention group, the results show that the BEW intake group significantly reduced (41%) the blood glucose level of hyperglycemic db/db mice. The intervention group demonstrated a significant increase in blood insulin compared with the control group. Significant histological differences were noted between db/db mice and their control mice. The results clearly show that BEW treatment improved hyperglycemia in obese diabetic db/db mice and improved their glucose processing capacity [94].

Another study recruited 30 subjects diagnosed with T2DM. Participants were randomly divided into two groups: a treatment group who received BEW with pH 9 (*n* = 15), and a control group who received a placebo (ordinary mineral water) with pH 7 (*n* = 15). Compared with the control group, the fasting blood glucose (ΔFBG) decreased more in the treatment group (19.4 ± 1.68 mg/dL vs. 14.3 ± 3.64 mg/dL).

It is strongly recommended to use BEW and walking to improve oxidative stress and inflammation. However, there is a lack of information on the combination of the two to reduce inflammation and oxidative stress. In another study, 81 eligible patients with type 2 diabetes (T2DM) were randomly assigned to four groups through a single-blind method: the BEW intervention group (*n* = 20, consumed 2 L/day of BEW), the regular walking intervention group (*n* = 20, instructed to walk 150 min per week), the combined BEW and regular walking intervention group (*n* = 20), and the control group (*n* = 21). The results show that after 8 weeks of intervention, compared with the control group, participants in the conventional walking and AEW combined intervention group displayed a synergistic improvement in FBG, AOPP, AGEs, MDA, NLR, and WBC. The possible mechanism of this synergistic effect may be the influence of antioxidants [95].

### 7.2. Anti-Obesity

Obesity has reached the level of a global pandemic. Obesity is a major public health problem and increases the risk of certain chronic diseases, such as cardiovascular disease, hypertension, type 2 diabetes and fatty liver [96]. Excessive fat intake will progress to obesity [97], which is characterized by abnormally increased adipose tissue mass in the adipose tissue and liver, an imbalance in the adipokine level and an imbalance between pro-inflammatory and anti-inflammatory cytokines [98].

The relationship between BEW (pH = 9.5 ± 0.3, ORP = −325.0 ± 20.5 mV) and obesity is unclear. Studies divided five-week-old C57BL/6 mice in which obesity was induced by feeding with a high-fat diet into three groups—a normal group, which were fed with normal-fat diet (10% fat) and tap water (NC+ TW), a control group fed with a high-fat diet (45% fat) and tap water (HFD+ TW), and an experimental group fed with a high-fat diet and BEW (HFD+ BEW) for 12 weeks. The degree of adiposity and diet-induced obesity (DIO)-associated parameters were assessed. After 7 weeks of intake of BEW, the mice in the experimental group had a significantly reduced body weight compared with the control group, and also displayed reduced epididymal fat weight. To determine the effect of BEW on liver cholesterol metabolism, another study measured the mRNA expression of hepatic CYP7A1 and HMG-CoA reductase, which are involved in lipid metabolism and cholesterol homeostasis. The results show that the experimental group demonstrated significantly increased liver cholesterol 7α-hydroxylase (CYP7A1) gene expression compared with the control group. The results indicate that BEW intake inhibits the progression of HFD-induced obesity by ameliorating adiposity, regulating adipokines and inflammatory cytokine level, and influencing cholesterol homeostasis in the liver [6].

### 7.3. The Effects on the Gut Microbiota

The large bowel is home to a complex and diverse microbial community, which plays an important role in health through a symbiotic relationship with the host. The main function of microbiota in the intestinal lumen is fermenting dietary fiber and resistant starch to produce short-chain fatty acids (SCFA). SCFAs play important physiological roles in the colonic mucosa, including the stimulation of mucus secretion, increased motility, and the absorption of sodium and water [99]. In addition, SCFAs can regulate systemic physiological and pathophysiological events, regulate the sympathetic nervous system, control body energy utilization, and differentiate immune cells [100]. In summary, the changes in microbiota composition and the changes in SCFA content play a key role in host health and disease.

The effects of BEW on the mouse (5-week-old male C57BL/6N mice) intestinal environment, including microbial composition and short-chain fatty acid content, were investigated. The experiment was divided into two groups, a control group and an experimental group, given tap water (pH = 6.83) or BEW (pH = 9.90) as drinking water for four weeks. The results show that the experimental group had significantly lower serum LDL-C levels and ALT activity. Regarding SCFA contents, the experimental group produced significantly more propionic acid, isobutyric acid and isovaleric acid than the control group, but no differences were found in succinic acid, lactic acid, formic acid, acetic acid, butyric acid and valeric acid. Additionally, 16S rRNA gene sequencing analysis results show that the relative abundance of the 20 taxa of experimental group was significantly different. Although the definite role of BEW administered to mice intestinal microbes was unknown, the results show that BEW may affect the composition of intestinal microbes and may be beneficial to health in terms of cholesterol metabolism and liver protection [7].

### 7.4. Anti-Tumor

It has been demonstrated that EW has reactive oxygen species (ROS) scavenging activity and can be used to treat oxidative stress-related diseases [101,102]. Regarding the anti-cancer effect of EW, it has been reported to down-regulate VEGF gene transcription and protein secretion by inactivating ERK, thereby reducing tumor blood vessel generation [8]. It was reported that NEW (pH = 6.6–7.8) can inhibit the colony formation and cell proliferation of human tongue squamous cell carcinoma-derived cell line HSC-4, but did not inhibit normal human tongue epithelioid cells DOK, and also inhibited the proliferation and invasion of human fibrosarcoma HT-1080 cells [103]. EW treatment of MCF-7, MDA-MB-453, and mouse (TUBO) breast cancer cells inhibited cell survival in a time-dependent manner. EW decreased ErbB2/*neu* expression and impaired pERK1/ERK2 and AKT phosphorylation in breast cancer cells. Overall, these results indicate that EW has a potentially beneficial effect in inhibiting the growth of cancer cells. Further research is needed to investigate the potential use of EW in breast cancer treatment [104].

## 8. Conclusions and Perspective

EW is mainly divided into two different types, acidic and alkaline. AEW has a very powerful effect as a disinfectant, and BEW has great potential for human health, such as diabetes, obesity, and cancer. Accumulating evidence was published subsequently. The effect of EW is also affected by many factors, such as pH value, ACC value, ORP value and storage time. Changes in these factors will affect the mechanism of action and reduce the effect. Although the mechanisms of EW as a disinfectant are very complete, as the clinical application mechanism is relatively limited, further discussion and research are needed. In addition, there are also some technologies that when combined with EW have an additive effect, and it is also worth studying these in the future.

## Figures and Tables

**Figure 1 molecules-27-01222-f001:**
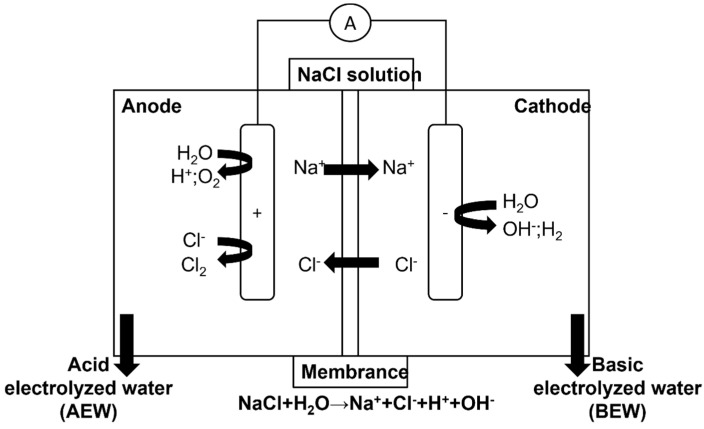
Schematic diagram of basic and acidic electrolyzed water.

**Table 1 molecules-27-01222-t001:** The characteristics of EW.

Type of EW	Solution	pH	ORP (mV)	ACC
Acid electrolyzed water	NaCl water (<0.2%)	2–2.7	> 110	20–60
Weak acid electrolyzed water	HCl water (2–6%) The mixture water of NaCl and HCl	5–6.5	850	10–30
Neutral electrolyzed water	NaCl or HCl	7–8	750–900	30–200
Basic electrolyzed water	NaCl water	10–13	−800–900	80–100

**Table 2 molecules-27-01222-t002:** Summary of EW’s antimicrobial activity.

Application	Microorganism	Log Reduction (CFU/Unit)	pH Value	ORP (mV)	ACC (mg/L)	References
Vitro	*E*. *coli* O157:H7	>7	2.36	1153	86.3	[26]
	*S. enteritidis*		2.48	1153	83.5	
	*L. monocytogenes*		2.63	1160	43.0	
Vitro	*Staphylococcus aureus*	5.8	6.1	893.5	30	[30]
Eggs shell	*L. monocytogenes*	4.01	2.5	1117	41	[47]
	*S. enteritidis*	3.81				
Tilapia	*E*. *coli*	1.68	2.47	1159	120	[48]
	*V. parahaemolyticus*	3.84				
Live clams and mussels	*E. coli* O104:H4	0.7–1.1	3.1	1150	20	[49]
	*L. monocytogenes*	0.6–0.9				
Salmon fillets	*L. monocytogenes*	0.75	2.6	1140	65	[50]
Biofilm	*E. coli*	0.7	2.94	1087	48.3	[48]
Laying-hen house	Airborne bacterial concentration	0.7	5.8–6.2	NA	150–250	[51]
Fresh fruits	*E. coli*	2.28	5.42	818	30	[52]
	*L. monocytogenes*					
Live clams and mussels	*E. coli* O104:H4	1.4–1.7	3.55	950	10	[49]
	*L. monocytogenes*	1.0–1.6				
Suspension	*E. coli*	6.02	6.1	863.5	30	[53]
	*S. aureus*					
Squid	Total bacterial counts	1.46	6.48	882	25	[54]

**Table 3 molecules-27-01222-t003:** Combinations of AEW and WAEW with organic acids.

Combined Treatments	Application	Microorganism	Log Reduction (CFU/Unit)	References
AEW + 1% citric acid	Lettuce	*L. monocytogenes*	2.6–3.7	[39]
WAEW + fumaric acid	Fresh beef	*Staphylococcus aureus*	0.8–1.6	[52]
		*L. monocytogenes*	1.17–2.12	
		*E*. *coli* O157:H7	1.15–2.01	
		*Salmonella* Typhimurium	1.15–1.81	
WAEW + fumaric acid	Fresh pork	*E*. *coli* O157:H7	2.59	[57]
		*L. monocytogenes*	2.69	
		*Staphylococcus aureus*	2.38	
		*Salmonella* Typhimurium	2.99	

**Table 4 molecules-27-01222-t004:** Combinations of AEW and WAEW with ultrasound.

Combined Treatments	Application	Microorganism	Log Reduction (CFU/Unit)	References
AEW + US	Strawberries	*E*. *coli* O157:H7	0.7–1.9	[60]
AEW + US	Suspension	*Salmonella* spp.	4.8	[61]
WAEW + US	Fresh fruits	Total aerobic bacteria	1.29–1.77	[59]
WAEW + US + mild heat	Fresh-cut bell pepper	*L. monocytogenes*	3.0	[62]
		*Salmonella* Typhimurium	3.0	

**Table 5 molecules-27-01222-t005:** Combinations of AEW and WAEW with thermal processing.

Combined Treatments	Application	Microorganism	Log Reduction (CFU/Unit)	References
AEW at 50 °C	Lettuce	*E*. *coli* O157:H7	3.0	[64]
		*Salmonella* spp.	4.0	
AEW at 65 °C	Atlantic salmon	*L. monocytogenes*	2.4	[63]
WAEW at 45 °C	Carrots	Aerobic bacteria	2.2	[65]

**Table 6 molecules-27-01222-t006:** Combinations of AEW and WAEW with UVC-LED.

Combined Treatments	Application	Microorganism	Log Reduction (CFU/Unit)	References
AEW + UV + US	Raw salmon	*L. monocytogenes*	0.64	[50]
WAEW + UVC-LED	Lettuce	*Salmonella* spp.	2.56–2.97	[69]
WAEW + UVC-LED	Coriander	*E*. *coli* O157:H7 *Salmonella* spp.	NA	[68]

**Table 7 molecules-27-01222-t007:** Summary of AEW and WAEW antifungal effects.

Application	Microorganism	Results/Log Reduction (CFU/Unit)	pH Value	ORP (mV)	ACC (mg/L)	References
In vitro	*A. flavus*	1.33–1.51	2.69	1125.7	90.3	[37]
In vitro	*Aspergillus* sp. *Paecilomyces* sp.	<3	2.65–2.76	1120–1188	60–121	[71]
Suspension	*A. flavus* *A. niger*	100%	2.8–2.9	1071–1079	54–56	[73]
Suspension	*Penicillum expansum*	1.61–2.73 4.37–4.85	NA	NA	10.1 59.6	[74]
In vitro	*A. flavus* conidia	ND	2.3–2.7	1045–1110	28–6.1	[70]
Tangerine	*Penicillium digitatum*	100%	3.9	NA	102	[72]
Pineapple	*Fusarium* sp.	60%	NA	NA	100 200 300	[75]

**Table 8 molecules-27-01222-t008:** Summary of AEW and WAEW’s antiviral effects.

Application	Microorganism	Results/Log Reduction (CFU/Unit)	pH Value	ORP (mV)	ACC (mg/L)	References
In vitro	avian influenza viruses (H5N1, H9N)	>5	1.5–2.5	NA	19–120	[82]
In vitro	avian influenza viruses (H5N1, H9N)	>5	6.4–7.4	NA	39–340	[82]
In vitro	Norovirus (MNV-1)	5.52	2.76	1138	10–60	[77]
	Hepatitis A virus (HAV)	5.19				
In vitro	Norovirus (MNV-1)	5.35	6.4	925	10–60	[77]
	Hepatitis A virus (HAV)	5.09				
In vitro	SARS-CoV-2 (JPN/TY/WK-521 strain)	≥4.25 log_10_ TCID_50_/mL	2.5	NA	66–109	[78]
In vitro	SARS-CoV-2	≥99.99% inactivation	2.5	NA	74	[83]
Blueberries	Norovirus (MNV-1) Bacteriophage MS2 Bovine rotavirus (boRV) Hepatitis A virus (HAV)	>4	5.0–8.5	700–900	200	[84]

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
