# Peer review of "Electrolyzed Water and Its Pharmacological Activities: A Mini-Review"

_molecules, 2022, doi:10.3390/molecules27041222_

Round 1

Reviewer 1 Report

I suggest a major revision before it can be acceptable for publication. The following suggestions should be considered:

(1) The English should be further polished to make the manuscript readable.

(2) The introduction section need to be rewritten. L25-31 seems has no relationship with EM. 

(3) In the introduction section, the authors should provide the highlights of this review and explain what new knowledge this paper can provide to the readers.

Author Response

Dear Reviewer, thank you for kind suggestion and comments. All the questions are answered as the followings.

  • The English should be further polished to make the manuscript readable.

Ans:

Thank you for good comments. All the manuscript have been revised and presented in the readable one.

  • The introduction section need to be rewritten. L25-31 seems has no relationship with EW.

Ans:

Thank you for kind advice. The introduction section has been rewritten according to your suggestion.

  • In the introduction section, the authors should provide the highlights of this review and explain what new knowledge this paper can provide to the readers.

Ans:

Thank you for good suggestions. The highlights of this review have been added: [Therefore, the use of EW is a good option. EW has the advantages of low cost and low harm to the human body. AEW exhibits strong bactericidal ability and antiviral potential, BEW has multifunction to promote human health. The purpose of this review is to introduce the latest development of electrolyzed water and provide new perspective in the clinical fields. Also clearly  introduces EW, including physicochemical properties, history, production, antimicrobial effects, and its prevention on diseases and health promotion.]

Reviewer 2 Report

Authors must also mention measures to be taken to make EW easily accessible to people. How to make it common. Authors have pointed out benefits of EW nicely.  

Author Response

Dear Reviewer,

Thank you for kind suggestion and comments. All the questions are answered as the following.

  • Authors must also mention measures to be taken to make EW easily accessible to people. How to make it common. Authors have pointed out benefits of EW nicely.

Ans:

Thank you for good comments. We revised it as: [E EW is a chlorine-based disinfectant that can be relatively easily prepared on-site by electrolyzing a solution of pure table salt (sodium chloride, NaCl) using one of many commercially available electrolysis equipment (106). With the latest developments in technology, industry has improved this technology to increase the effectiveness of electronic warfare. Since 2010, many innovative companies have been emerged in the market, and the electronic warfare generators can also be used by individuals and small businesses (107,108).]

  • Authors must also mention measures to be taken to make EW easily accessible to people. How to make it common. Authors have pointed out benefits of EW nicely.

Ans:

Thank you for good comments. We revised it as: [E EW is a chlorine-based disinfectant that can be relatively easily prepared on-site by electrolyzing a solution of pure table salt (sodium chloride, NaCl) using one of many commercially available electrolysis equipment (106). With the latest developments in technology, industry has improved this technology to increase the effectiveness of electronic warfare. Since 2010, many innovative companies have been emerged in the market, and the electronic warfare generators can also be used by individuals and small businesses (107,108).]

Round 2

Reviewer 1 Report

The paper can be considered for publication after moderate English polishment.